



# The role of Arctic sea ice loss in projected polar vortex changes

Marlene Kretschmer[1], Giuseppe Zappa[1,2] and Theodore G. Shepherd[1]

[1]Department of Meteorology, University of Reading, Reading, UK

[2]Istituto di Scienze dell'Atmosfera e del Clima, Consiglio Nazionale delle Ricerche, Bologna, Italy

*Correspondence to*: Marlene Kretschmer (m.j.a.kretschmer@reading.ac.uk)

**Abstract.** The Northern Hemisphere stratospheric polar vortex (SPV) plays a key role for mid-latitude weather and climate. However, in what way the SPV will respond to global warming is not clear, with climate models disagreeing on the sign and magnitude of projected SPV strength change. Here we address the role of Barents and Kara (BK) sea ice loss in this. We provide evidence for a non-linear response of the SPV to global mean temperature change, dependent on the time the BK Seas become ice-

free. Using a causal network approach, we demonstrate that climate models show some partial support for the previously proposed link between low BK sea ice in autumn and a weakened winter SPV, but that this effect is plausibly very small relative to internal variability. Yet, given the expected dramatic decrease of sea ice in the future, a small causal effect can explain all of the projected ensemble-mean SPV weakening, approximately one-half of the ensemble spread at the middle of the 21st century, and one-third of the spread at the end of the century. Finally, we note that most models have unrealistic amounts of BK sea ice, meaning that

their SPV response to ice loss is unrealistic. Bias-adjusting for this effect leads to pronounced differences in SPV response of individual models at both ends of the spectrum, but has no strong consequences for the overall ensemble mean and spread. Overall, our results indicate the importance of exploring all plausible implications of a changing Arctic for regional climate risk assessments.

## 1 Introduction

The stratospheric polar vortex (SPV), a band of fast-blowing westerlies forming during boreal winter, is a central component of the Northern Hemisphere atmospheric circulation (Waugh et al., 2016). Variability in vortex strength is not only linked to stratospheric ozone concentrations, but due to downward coupling to the troposphere also strongly affects mid-latitude weather. In particular, extreme states of the SPV are known to influence the phase and persistence of the North Atlantic Oscillation and associated storm tracks and weather regimes (Baldwin & Dunkerton, 2001; Kidston et al., 2015).

Understanding potential changes of the SPV in response to global warming is therefore of huge scientific and societal relevance. If the vortex strengthens, for instance, Mediterranean precipitation is expected to strongly decrease while days of extreme storminess in northern Europe are expected to increase (Simpson et al., 2018; Zappa et al., 2017). Conversely, if the vortex weakens, the pace of Mediterranean drying is likely more moderate and changes in storminess less pronounced.

However, in what way the SPV will respond to a warming climate in the future is highly uncertain. While the multi-model ensemble
mean from Phase 5 of the Coupled Model Intercomparison Project (CMIP5) under the RCP8.5 scenario suggests a moderate weakening at the end of the 21st century, there is huge inter-model spread and no agreement on the sign of change (Manzini et al., 2014; Wu et al., 2019). Also for the next generation of CMIP models, this remains an issue (Ayarzagüena et al., 2020).

This spread is not just attributable to different vertical resolutions and model lids and has been speculated to depend on distinct wave parametrizations and differently represented dynamical processes (Karpechko et al., 2017; Sigmond et al., 2010; Wu et al.,
2019). Several potential mechanisms influencing SPV strength have been reported in this context, such as El Nino Southern



Oscillation (ENSO) or high-latitude blocking (Domeisen et al., 2018; Martius et al., 2009; Nishii et al., 2010; Peings, 2019), but their relative importance and their role in a changing climate are not well understood (Shepherd, 2014; de Vries et al., 2013). Overall, the future polar vortex change remains thus completely undefined.

Recently, Manzini et al. (2018) reported a non-linear response of the SPV to global mean warming in a single-model large

ensemble, and hypothesized it to be related to Arctic sea ice loss. More precisely, the SPV was proposed to weaken as long as Barents and Kara sea ice concentrations (BK-SIC) decreased, but to strengthen again once the BK Seas were ice-free (Manzini et al., 2018).

Motivated by their results, we here assess the role of BK sea ice loss for future SPV changes in the CMIP5 ensemble. Whilst the question of whether Arctic sea ice loss contributed to the recent episode of weak vortex events (and associated cold-air outbreaks

over Eurasia) remains an active area of research (Kim et al., 2014; Kretschmer et al., 2016a, 2018; McCusker et al., 2016; Seviour, 2017; Sun et al., 2016), the potential of decreasing BK-SIC to weaken the SPV has been robustly shown in various targeted model experiments (Blackport et al., 2017; Hoshi et al., 2017; McKenna et al., 2017; Nakamura et al., 2016; Screen, 2017a, 2017b; Sun et al., 2015; Zhang et al., 2017, 2018). Nevertheless, possible implications of future sea ice loss have so far been only rarely studied (Sun et al., 2015), but instead, findings on conflicting model and observational results have dominated the scientific debate (Cohen

et al., 2020;  Screen et al., 2018), stressing a likely small and statistically insignificant influence of sea ice on SPV strength compared to natural variability (Blackport et al., 2019; Blackport & Screen, 2020; Sun et al., 2016). As the decline of Arctic sea ice in a warming climate is certain (IPCC, 2014; Notz & Stroeve, 2018) understanding potential impacts on future SPV strength is crucial and forms the aim of the present analysis.

**2 Data**

We use monthly outputs from 35 CMIP5 models (see captions in Fig. 3), for which data are available for our purposes. For each model, the historical (1900-2005) and RCP8.5 (2005-2099) simulations from the same ensemble member are concatenated to produce a continuous time-record in the analysed fields. All available ensemble members are considered separately when analysing time-series and their number per model is reported on the x-axis of Fig 3. For all other analyses the mean over the available

ensemble members per model is calculated first.

For observations of sea ice concentration we use the latest version (HadISST.2.2.0.0) of version 2 of the Hadley Centre Sea Ice and Sea Surface Temperature data set HadISST.2 (Titchner & Rayner, 2014). Note that this sea ice product gives a rather conservative estimate of monthly sea ice, in particular having higher mean concentrations compared to HadISST.1. For all other variables, we use monthly means of ERA5 reanalysis data as a measure for observations (Hersbach et al., 2020). Analysis for the

observations are performed over the period 1979-2018.

Time-series are constructed by area-averaging over the respective regions, whereby we include Barents and Kara sea ice concentrations ("BK-SIC") over the region 65-85°N and 10-100°E (Screen, 2017a), sea level pressure over the Ural Mountains region ("Ural-SLP")  45-70°N and 40°-85°E (Kretschmer et al., 2016b) and over the North Pacific ("NP-SLP") 30-65°N and 160-220°E (Trenberth & Hurrell, 1994). As a proxy for vertical wave activity flux, we compute poleward eddy heat flux ("vT") at 100

hPa averaged over 50-80°N (Hoshi et al., 2017). More precisely, the zonal-mean deviations of the monthly-mean meridional winds and temperature at 100hPa at each grid-point are first multiplied and then the spatial mean is calculated. To describe the



stratospheric polar vortex ("SPV"), we follow Wu et al. (2019) and compute the average zonal wind velocity over 60-75°N but at 20 hPa instead of 10 hPa.

## 3 Methods

### 3.1 Changes as a function of global mean warming and BK-SIC loss

Anomalies of seasonal-mean SPV (JFM), BK-SIC (OND) and all-year global mean temperature (T) are calculated by subtracting the mean over the reference period 1960-1989. Similar to Manzini et al. (2018), we then calculate a 15 year-mean moving average of global mean temperature change ($\Delta T$) and include the last 14 years from the historical simulations to calculate the average over the first window. The last year of each window represents this average, that is, the year 2006 represents the mean over 1992-2006. A reached global warming level is defined as the first time this 15-year average is equal or larger than a certain threshold level. The SPV change for a given warming level is then calculated as the 30-year average before this warming level was reached. For example, if a warming of 5 K was reached in the year 2099 (i.e. the global mean temperature change averaged over 2085-2099 exceeds 5K), SPV change is calculated over the period from 2070-2099. We proceed equivalently when plotting BK-SIC change as a function of temperature change, and when plotting SPV change as a function of BK-SIC change.

### 3.2 Estimating the timing of an ice-free BK

The year the BK Seas become ice-free is here defined as the first year that projected BK-SIC anomalies (relative to the reference period 1960-1989) fall below 5% of the observational mean of 0.51 (calculated over the same reference period). This refers to the fraction of BK-SIC being lower than 0.026, i.e. less than 2.6% of the BK Seas are covered with sea ice. For models that are not ice-free before the end of the 21[st] century, we calculate the expected year they become ice free by fitting a linear trend line over the years 2006-2099. The year this trend line is below 0.026 is then defined as the year the model´s BK Seas are expected to become ice-free (see Fig. 1d).

## 4 Results

### 4.1 The non-linear response of the polar vortex

To test for evidence of a non-linear SPV response related to sea ice loss in the CMIP5 models, we plot the projected SPV change in January-March (JFM) and BK-SIC change in October-December (OND) for different levels of global mean warming in the RCP8.5 high-emissions scenario (Fig. 1a, b). We show the evolution of each model (grey lines) as well as the multi-model mean (blue lines), with darker shades of blue indicating means over the subset of models with stronger warming at the end of the 21[st] century.

The ensemble-mean SPV weakens by up to approximately 2 m/s for 2.5 K warming, and strengthens slightly afterwards for models reaching 5 K warming at the end of the century (see dark blue line), or remains constant (lighter blue lines). This coincides with a





flattening of the multi-model mean BK-SIC change, indicating that the BK Seas have become ice-free in several models (see dark blue line and upper bounds of the ensemble spread in Fig. 1b). In contrast, when plotting SPV change as a function of BK-SIC

change, we find it to be approximately linear, with most models showing a weakening of the SPV while BK-SIC decreases (Fig. 1c).

Interestingly, not only does the maximum temperature change vary across models as a result of different climate sensitivities, but also the amount of BK-SIC loss varies substantially, because of different BK-SIC climatologies. In fact, the timing of an ice-free BK (see methods) can be well predicted from the model´s BK-SIC climatology divided by the projected global mean warming at

the end of the 21$^{st}$ century (r = .82, Fig. 1d). For 66% of the models the BK Seas are ice-free in OND before the year 2100. This includes in particular all models but one model with below-average sea ice conditions compared to observations. Models with a lot of initial sea ice, in contrast, are on average ice-free later.

To account for this spread regarding the timing of BK-SIC being gone, we next show the 30-year running mean SPV change, aligned and normalized by the reference to the year the BK Seas become ice-free (Fig. 1e). Thus, by construction, all time-series

have value 1 at year 0 (the year BK-SIC is gone). For consistency with Fig. 1a-c, the SPV change is evaluated relative to the 1960-1989 period, but our results are not sensitive to the chosen start period. To aid visualization, start and end values are highlighted with dots. Before the BK Seas are ice-free (grey lines and dots), values above 1 thus indicate a weakening with time of the SPV, while they imply strengthening afterwards (blue lines and dots). Up to the time the BK Seas are ice-free, the SPV weakens in two-thirds of the models. Afterwards, only a few models show further weakening, while most indicate a strengthening SPV (values

above one) or no further change (values close to one). Thus, there is an indication of a weakening signal of the SPV in CMIP5 models up to the point where BK-SIC is gone, with the response switching sign thereafter.

This difference in SPV change before and after the BK Seas are ice-free is further shown in a box and whiskers plot (Fig. 1f). As we compare changes over different time intervals, we divide the SPV change by the model´s global mean temperature change over the considered time-span. The CMIP5 ensemble shows a robust weakening SPV signal before BK-SIC is gone with most of the

inter-quartile range being below zero (grey box plot). For those models for which sea ice is gone before the end of the 21$^{st}$ century, the SPV strengthens on average

Overall, consistent with Manzini et al.'s (2018) single-model results, Fig. 1 thus suggests a non-linear response of the SPV in CMIP5 models to global mean warming, dependent on the timing of when the BK Seas become ice-free.

**4.2 Potential confounding factors**

In the following we aim to understand the contribution of BK-SIC change to SPV change in more detail. This is challenging given a fully coupled climate system with several, likely competing effects which might both enforce or dilute the signal of interest. In this context, simple regression or correlation analyses have been questioned with respect to their causal interpretation, as they exhibit several limitations (Blackport et al., 2019; Kretschmer et al., 2016a).

For example, common drivers can spuriously increase the regression strength or may even, if the influence is of opposite sign, dilute the true relationship between two processes (Kretschmer et al., 2016b). Further, auto-correlation of a time-series, which is characteristic of sub-seasonal Arctic sea ice concentrations or polar vortex strength, inflates the correlation strength, potentially




suggesting a false link between two processes (Runge et al., 2014). Thus, to estimate the influence of BK-SIC on SPV, one has to
control for such factors.

To do this, we need to assume a causal model of the underlying processes, here shown in the form of a graphical network (Fig. 2).
Nodes represent different sub-seasonal processes and the arrows indicate causal relationships between them, with arrows self-
connecting a node representing auto-dependence of that process. This network can be interpreted as our attempt to summarize the
large body of literature on the topic in the most parsimonious way, recognizing it as being prone to subjective judgement and only
representing a reduced model of the underlying truth.

A reduction in Barents and Kara sea ice concentrations in autumn and early winter ("BK-SIC") is assumed to increase turbulent
heat-flux in this region, leading to enhanced sea level pressure over the Ural Mountain region ("Ural-SLP"), as shown by several
studies (Kim et al., 2014; Kug et al., 2015). Via constructive interference with the climatological stationary wave this enhances the
vertical wave activity flux ("vT") into the stratosphere causing a weakening of the vortex ("SPV") in winter (Kretschmer et al.,
2016a; Peings, 2019). However, Ural-SLP also affects BK-SIC (Blackport et al., 2019; Tyrlis et al., 2019), making it hard to isolate
the signal emerging from sea ice alone. Further, tropical Pacific variability, e.g. in the form of El Niño Southern Oscillation or the
Madden-Julian Oscillation ("ENSO/MJO"), can affect vT and thus the SPV via altered sea level pressure anomalies over the North
Pacific ("NP-SLP") (Domeisen et al., 2018). Furthermore, sea ice decline in the North Pacific ("NP-SIC") has also been causally
linked to lower NP-SLP and thus a strengthened SPV (Kug et al., 2015). As NP-SLP can also affect Ural-SLP via Rossby wave
propagation, it confounds the analysis of the Arctic-Stratosphere pathway (Jiménez-Esteve et al., 2018; Warner et al., 2020).

Note that downward stratosphere-troposphere coupling is not included here as we consider tropospheric processes in autumn and
early winter only, with the downward links, e.g. from SPV to Ural-SLP or BK-SIC, expected in mid- and late winter (Kidston et
al., 2015; Smith et al., 2018). That is because our interest here is in understanding the response of SPV to global warming.

### 4.3 Estimating the causal effect of BK-SIC on SPV

Making our assumptions of the underlying causal structure explicit has the advantage that it guides further statistical analyses. We
next try to quantify the (indirect) influence of autumn BK-SIC on winter SPV in the historical simulations. We recognize that an
exact quantification is probably not possible due to the large internal variability, including a documented intermittency of the
Arctic-Stratospheric pathway (Kolstad & Screen, 2019; Siew et al., 2020), and uncertainties regarding the involved time-lag
(Blackport & Screen, 2019; García-Serrano et al., 2017). Our aim, therefore, is to come up with a *plausible* estimate of the mean
causal effect.

To achieve this, it is according to causal inference theory sufficient to "block" all confounding processes (Pearl, 2013), i.e.
processes that influence both autumn BK-SIC and winter SPV. Assuming linear dependence, this can be done by regressing the





seasonal-mean winter (JFM) SPV on late autumn (OND) BK-SIC and, to control for confounding, additionally regress on autumn (SON) Ural-SLP., i.e.

$$SPV_{JFM} = a\ BK\text{-}SIC_{OND} + b\ Ural\text{-}SLP_{SON}$$

whereby a is interpreted as the mean causal effect of autumn BK-SIC on winter SPV. For consistency with Fig. 1, we used seasonal-mean data and therefore do not control for autocorrelation. We explicitly do not control for winter (DJF) Ural-SLP and vT as the influence of BK-SIC is assumed to be mediated by these variables (Fig. 2) and we would thus regress out exactly the pathway we aim to measure. Note further that the confounding effect of NP-SLP is mediated via Ural-SLP and thus accounted for. Yet, even

when including autumn NP-SLP in the regression, our results are only marginally affected.

To account for sampling uncertainties and facilitate comparison with observations, we calculate the regression over different 39-year-long moving windows over the historical simulations from 1900-2005, resulting in 67 partly-overlapping windows overall. As we compare models with different SPV and BK-SIC variabilities, the regressions were performed over standardized time-series by first subtracting the mean and then dividing over the standard deviation for each season. Both mean and standard deviations are

calculated over the considered 39-year time window. Further, linear trends are removed by fitting a regression slope over the time-series over each window.

Fig. 3a shows the spread of the regression parameter a for different models and time-windows (left panel), indicating large intra-model and inter-model spread. Yet, as expected, most models have a positive mean causal effect (right panel Fig. 3a) with a median of 0.035. The histogram of all link strengths (middle panel of Fig. 3a) shows a bell-shaped distribution with a positive mean of

0.052, meaning that, on average, a change by one standard deviation ($\sigma$) in BK-SIC leads to a 0.052 $\sigma$ change in SPV.

We get slightly higher regression coefficients when averaging BK-SIC over ND and SPV over JF only (not shown). Results are also similar when using monthly time-series and additionally controlling for autocorrelation of BK-SIC. Overall, albeit the signal being weak, there is thus evidence for less BK-SIC in autumn causing a weakening of the SPV in winter, under the premise of the causal model being true.

Note that the causal effect for the observations is as high as 0.38, being on the outer tail of all computed link strengths (blue cross in middle panel in Fig. 3a). Previous studies suggested models to systematically underestimate the effect emerging from BK-SIC (Cohen et al., 2020). On the other hand, this potential discrepancy between models and observations was also attributed to a relatively active stratospheric pathway over recent years (Siew et al., 2020), therefore not being representative of the actual, likely





much lower link strength. Our results could be interpreted in both ways but addressing this aspect lies outside the scope of the
present analysis.

### 4.4 Representation of the Arctic-Stratospheric pathway

In an attempt to better understand the inter-model spread, we further compute the link strengths of the assumed mediating processes.
This can be done by regressing each process Y on its ingoing links (Fig. 2) with the regression coefficients interpreted as the
respective link strength (Pearl, 2013).

For example, to estimate the effect of November BK-SIC on January Ural-SLP, we compute

$$\text{Ural-SLP}_J = a\,\text{BK-SIC}_N + b\,\text{NP-SLP}_D$$

with a denoting the causal effect of BK-SIC (Fig. 3b) on Ural-SLP, and b that of NP-SLP (Fig. 3c). Both effects are found to be
very weak on these monthly time-scales and can only be seen in roughly half of the model averages with no signal in the multi-
model median (right panels Fig. 3b, c). In a similar way, we also compute the influence of November Ural-SLP on December BK-
SIC (Fig. 3d), of December Ural-SLP and NP-SLP on January vT (Fig. 3e, f), and of January vT on February SPV (Fig. 3g).
Though the spread in link strength within and across models is again large, they are mostly of the expected sign and results are
robust when choosing different winter months.

The weak or missing mediated signal from BK-SIC to Ural-SLP illustrates a dilemma, frequently faced when studying the impact
of sea ice on mid-latitude circulation. On the one hand, if our Null-Hypothesis was the non-existence of such a link, we could not
reject it based on the presented results (avoidance of type-1 error). On the other hand, as failure to reject a hypothesis does not
prove the hypothesis, we cannot rule out the possibility of an influence of BK-SIC on SPV via Ural-SLP (avoidance of type-2
error). For example, our choices on the regional indices as well as the time-scales and time-lags might not be optimal, and/or be
model dependent hence watering down the mean signal.

Some models seem to be systematically underestimating the assumed Arctic-Stratospheric pathway. For example, model 16
("FGOALS-g2") is a notable outlier for the link from Ural-SLP to vT (Fig. 3e). However, as the sample sizes are small and the
analyses only represent proxies and "snapshots" of the studied links, there is no obvious justification for excluding models on this
basis to reduce the spread in future SPV projections. Doing this would require a more detailed analysis of the processes and time-
scales in the individual models.

Overall, we can thus neither prove nor disprove the representation of the individual chain of mediating links in the historical
simulations of the CMIP5 ensemble. However, rejecting our initial assumption of a causal link from BK-SIC to SPV would leave
us with the problem of explaining the non-linear SPV response to global warming, as presented in Fig. 1. Therefore, our approach
will now consist in exploring the implications for the SPV under climate change, assuming that a weak signal from BK-SIC to
SPV indeed exists as suggested by previous studies (De & Wu, 2019; Kim et al., 2014; Screen, 2017b; Zhang et al., 2018) and





supported by Fig. 3a. While this is important on its own, this approach will also enable us to provide some indirect support to the
       existence of the link itself.

**4.5 Implications for projections of SPV change**

       Addressing possible implications of future BK sea ice loss for SPV change seems particularly justified given an expected decrease
of sea ice under global warming (Notz & Stroeve, 2018), making it necessary to assess related risks (Sutton, 2019). Further, results
       presented in Fig. 1 indicate a role of BK-SIC for SPV change, which we try to understand.

       To do this, we test how well the projected BK-SIC changes (relative to 1960-1989) can explain the projected SPV changes across
       the RCP8.5 simulations, for different assumed standardized causal effects (ce) of 0.025, 0.05, and 0.1. These levels are motivated
       by the regression strength found over the historical period (Fig. 3a), representing plausible estimates of the mean causal effect of
BK-SIC on SPV. For example, 0.05 is about the ensemble-mean regression strength (0.052), and 0.025 is slightly below the
       ensemble median (0.035), and 0.1 is slightly below the upper quartile range (0.14). Using only one (standardized) effect for the
       whole ensemble further seems reasonable, as there is no strong evidence of models having very different behaviours (Fig. 3).

       To express ce in physical units and to account for different variability in the different models, we weight it by ratios of standard
       deviations ($\sigma$, calculated over the reference period 1960-1989) for each model m, i.e.

$$CE_m = ce \cdot \sigma_{SPV_m} \cdot \sigma_{BK\,SIC_m}^{-1}$$

       The distribution of these values for ce = 0.05 over the different models is shown in Fig. 4a. Note that seasonal-mean (OND) and
       individual monthly variability are comparable for most models. Only for model 5 ("BNU-ESM") and model 27 ("IPSL-CM5A-
       MR") does this not hold as these models have basically constant sea ice conditions in December. In the following these models are
       therefore excluded from the presented results, but including them does not change our main results.

We next show the scatter plots of predicted winter (JFM) SPV change based on autumn (OND) BK-SIC change versus the actual
       projected SPV change for mid-century (2040-2069, Fig. 4c) and end-of-century (2070-2099, Fig. 4d). For the former period, the
       statistically predicted and actual projected SPV change correlate significantly (r = 0.61, p<0.01 according to a two-sided Student´s
       t-test). This correlation is independent of the chosen causal effect strength but is a result of the correlation between BK-SIC and
       SPV change (r = 0.4), which increases after accounting for the different ratios of standard deviations between models.

Assuming a causal effect of 0.05, the prediction model explains 42% of the ensemble spread (measured in median absolute
       deviation, MAD) and the ensemble mean predicts a SPV change of -2.4 m/s, compared to an ensemble-mean projected SPV change
       of -1.8 m/s. Thus, the BK-SIC decrease can account for all of the projected mid-century SPV weakening (and beyond) and almost
       half of the ensemble spread. For the end-of-century prediction (Fig. 4d), the correlation is still statistically significant (p<0.01) but
       drops to 0.42 while the model still explains 38% of the ensemble spread and overestimates the mean change by about a factor of
two. For assumed causal effects of 0.025 and 0.1, the explained mean and variance halve and double, respectively.

       Building upon our initial analysis (Fig. 1), we hypothesize that the drop in correlation and overestimation of the mean SPV change
       between mid- and end-of-century is due to more models having ice-free BK Seas at the end of the century. This is tested by





calculating the correlation of predicted and projected SPV change over different moving windows within the 21$^{st}$ century (Fig. 4e).
For the 15 models that are ice-free late (after the year 2090), the predicted and projected SPV change correlate strongly (up to 0.8)

over the entire century. In contrast, models that are ice-free early (before 2090) only show a moderate correlation, which is dropping
and becoming negative in the second half of the century. This makes sense, as models that are ice-free late have more and longer
lasting BK-SIC (Fig. 1d) and thus the effect on SPV is more dominant. For models that are ice-free early, in contrast, the BK-SIC
effect on SPV change disappears once the BK-SIC forcing is gone.

Consistent with Fig. 1a, we find no correlation between SPV and global mean temperature change (turquoise line in Fig. 4f).

However, this may stem from the fact that models that are ice-free in the BK Seas early and late have correlations of opposite sign.
The moderate positive correlation (up to 0.5) of global mean temperature and SPV change for early ice-free models supports the
initial hypothesis that once BK-SIC is gone, it ceases to exert an effect, and the SPV strengthens in response to global mean
warming.

**4.6 The tug-of-war over future SPV change**

The estimated causal effect from BK-SIC to SPV allows us to decompose the projected SPV response to global warming (Figs.
5a, d) into a contribution from BK-SIC change (Figs. 5b, e) and all remaining factors (Figs. 5c, f). The latter is simply calculated
as the residuals, i.e. projected minus predicted SPV, and can be interpreted as the effect of global mean warming on the SPV
without the effect mediated via BK-SIC loss. We show the evolution of the three quantities both as a function of time (Figs. 5a-c)

and of global mean temperature change (Figs. 5d-f).

Assuming ce=0.05, the effect of BK-SIC loss on SPV change at the end of the century ranges from almost no change up to a
weakening of more than 10 m/s, with the ensemble-mean predicting a change of -3.4 m/s (Fig. 5b). Thus, even for a small
standardized causal effect of 0.05, the projected dramatic decrease of Arctic sea ice implies relatively large SPV changes (Figs.
5b, e). In fact, all of the projected ensemble-mean SPV change (thick lines in Figs. 5a, d) can be explained by BK-SIC decrease,

with the residual´s ensemble-mean being close to zero for the first half of the century and for global mean warming up to 2.5 K
(Figs. 5c, f). After the year 2060, the residual´s ensemble-mean even becomes positive and most ensemble members indicate a
strengthened SPV for a global mean temperature change above 2.5 K (Figs. 5c, f). Thus, if BK-SIC did not decrease, a moderate
strengthening of the SPV in response to global mean warming would be expected, which appears to be approximately linear (Fig.
5f).

For a doubled causal effect (ce = 0.1) of BK-SIC on SPV, the residual´s ensemble-mean would imply a strengthening of more than
5 m/s at the end of the century (dashed line in Fig. 5c). For a causal effect of only 0.025, in contrast, it basically implies no SPV
change over the 21$^{st}$ century in the residuals (dotted line in Fig. 5c).

In summary, Fig. 5 shows the opposing effects, often called a 'tug-of-war', on the SPV of climate change manifested by BK-SIC
decrease, and other effects not specified here. While BK-SIC decrease accounts for all the projected ensemble-mean SPV





weakening even for a small assumed ce of 0.025, the effect not mediated via BK-SIC ranges from no change (for ce = 0.025) to a pronounced strengthening (for ce = 0.1).

### 4.7 Bias-adjusted sea ice concentrations

Finally, we note that most CMIP5 models have too much initial BK-SIC compared to observations (Fig. 6a, see also Fig. 1d)
meaning that their SPV response to sea ice loss is too strong. The opposite holds for models with too little initial BK-SIC. We therefore include a simple bias-adjustment function to estimate the expected SPV change for more realistic sea ice conditions.

Fig. 6b indicates for each model the periods before and after the BK Seas have become ice-free as well as the periods where a bias adjustment is needed. For models with too much initial sea ice, the bias-adjustment function ramps up from zero when BK-SIC loss becomes unphysical, i.e. when it exceeds the observational BK-SIC, and stays constant as soon as the model´s BK Seas are
ice-free. The constant is the erroneous amount of BK-SIC loss in that model, multiplied by the specified causal effect $CE_m$ on SPV (see Fig. 4a). For models with too little initial sea ice, the bias-adjustment function ramps down from zero starting when the BK Seas are ice-free and remains constant after when the model would have been ice-free had it had a realistic initial BK-SIC (see methods).

Model 10 ("CMCC-CESM"), for example, exhibits large internal variability but shows a robust SPV weakening, reaching almost
10 m/s at the end of the century (light blue line in Fig. 6c). However, this model has almost twice too much BK-SIC compared to observations and thus the weakening response is likely overestimated. Bias-adjusting this effect, starting when the BK-SIC loss exceeds the physically possible amount (see blue square in Fig. 6c), roughly halves the projected weakening at the end of the century (thick blue line Fig. 6c) when assuming a causal effect of 0.05. An effect of 0.1 would imply almost no SPV change (dashed blue line in Fig. 6c) while that of 0.025 indicates a weakening of only around 7 m/s (dotted blue line in Fig. 6c).

In contrast, model 28 ("MIROC-ESM") is the model with the most pronounced SPV strengthening of approximately 5 m/s at the end of the century (thin red line in Fig. 6c). Interestingly, the strengthening only starts after the BK Seas have become ice-free (red square in Fig. 6c), consistent with the overall findings of this study. As the model has too little BK sea ice compared to observations, the projected pronounced strengthening is unrealistic. Adjusting for this bias leads to a strengthening of only about 2.5 m/s (thick red line in Fig. 6c) when assuming a causal effect of 0.05, and would even imply no change when assuming an effect
of 0.1 (dashed red line in Fig. 6c).

Despite these large effects for individual models, the ensemble-mean predicted SPV weakening from BK-SIC loss only marginally changes at the end of the century (see thick lines in Fig. 6d) while the spread is reduced by less than one-third for a causal effect of 0.05 (Fig. 6d). Bias-adjusting the initial sea ice conditions reduces the end-of-century ensemble-mean (with model 5 and model 27 being excluded) from -1.4 m/s to -1.1 m/s, and lifts the lower bound from -9.5 m/s to -9 m/s and the upper bound from 5.6 m/s





 to 5.5 m/s. Overall, the erroneous initial amount of BK-SIC in the models thus does not substantially affect the projected SPV change (Fig. 6e).

## 5. Discussion

Our study adds to the large body of literature addressing the overarching question of whether the Arctic *can*, *has* and *will* influence

mid-latitude weather and climate (Barnes & Screen, 2015). Several previous studies used the absence of a statistically significant signal to argue against an influence of sea ice (Blackport & Screen, 2020; Sun et al., 2016). However, absence of evidence is not evidence of absence (Shepherd, 2016). Moreover, if we reject the hypothesis of a causal influence, we are left with the puzzle of explaining the nonlinearity seen in Fig. 1, whereas a weak causal influence is consistent with a lack of statistical significance. We have shown that despite the signal emerging from BK-SIC being likely small relative to internal variability, and the proposed

mediating pathway not being robustly identified in the CMIP5 ensemble, the impacts for the future SPV could nonetheless be pronounced due to large loss of BK-SIC. In particular, assuming a weak influence of BK-SIC on SPV enables to explain both the non-linearity in the SPV response to warming (Fig. 5) and a large part of the inter-model spread in the model projections (Fig. 4). In turn, these findings provide some indirect evidence for the plausibility of this pathway. Given that declining Arctic sea ice is certain in a warming climate, we argue for putting more focus on the avoidance of type-2 errors (Shepherd, 2019), to fully address

the plausible range of regional climate impacts of Arctic sea-ice loss (Sutton, 2019).

Quantifying the influence of BK-SIC on SPV is difficult. One potential reason is the documented non-stationarity of the Arctic-Stratosphere pathway with several models exhibiting large decadal SPV variability (Kolstad & Screen, 2019; Siew et al., 2020). Another challenge we encountered is the choice of the relevant sea ice months, which seems to vary across models and might also change from year to year for a given model (Blackport & Screen, 2019; García-Serrano et al., 2017). For a fixed lag and month,

only approximately half of the models show an expected negative link from BK-SIC to Ural-SLP (Fig. 3b). On the other hand, using seasonal averages to estimate a net causal effect might have led to an underestimated causal effect in the models (Fig. 3a). An improved understanding of the timing of the Arctic-Stratosphere pathway will be necessary to achieve progress.

The uncertainty regarding future SPV change contributes to uncertainty about future mid-latitude weather and climate (Simpson et al., 2018; Zappa et al., 2017). Our results of a weakened SPV in response to BK-SIC decrease are overall consistent with a

reported poleward jet shift and negative NAO-like response to sea ice loss across the CMIP5 models (Blackport et al., 2017; De & Wu, 2019; Screen & Blackport, 2019; Zappa et al., 2018). In this context, it was further shown that models with sea-ice loss have a weakened SPV in late winter, whilst those without sea-ice loss have a strengthened SPV (see e.g. Fig. S5 of Zappa et al. 2018). Yet, how much the stratospheric pathway discussed in this study contributes to this compared to other Arctic-related mechanisms not involving the stratosphere remains an open question (Kretschmer et al., 2016a; Nakamura et al., 2016; Wu et al., 2016; Zhang

et al., 2018).

Identifying the processes causing a strengthening of the SPV is beyond the scope of the present study but is relevant to understand future polar vortex change. Changes in both vertical as well as horizontal wave activity propagation might play a role (Wu et al.,

2019). For example, a deepened Aleutian Low, as favoured by decreasing sea ice in the Pacific sector, might contribute to such a strengthening (Hu et al., 2018; McKenna et al., 2017; Nishii et al., 2010) (as suggested here in Fig. 3f).

More generally, this study shows the benefits of a causal network approach to identify and quantify teleconnection signals in multi-model ensembles. Making assumptions of the underlying causal model explicit transforms domain knowledge into mathematical testable objects and can guide the statistical analysis. Testing different hypotheses in this way is thus a logical next step to further constrain future SPV changes.

**6. Summary & Conclusion**

We have provided evidence for a non-linear response of the SPV to global mean warming, suggested to result from a weakening caused by sea ice loss in the BK Seas and opposing effects (not specified in this study) which dominate the SPV response once the BK Seas are ice-free. The timing of the latter varies substantially across models, as a result of different sea-ice climatologies and different warming rates.

A plausible quantification of this Arctic-Stratosphere teleconnection in the historical simulations resulted in a standardized causal effect in the range of only about 0.05, which helps explain why it is difficult to detect. Yet, the implications of such a small causal effect for future SPV projections in the RCP8.5 scenario are notable due to the expected dramatic shrinking of Arctic sea ice: BK-SIC change can explain all of the projected ensemble-mean SPV weakening and up to almost half of the ensemble spread over the 21st century.

We finally noted that most models include unrealistic sea ice conditions compared to observations and thus also their SPV response to sea-ice loss is unrealistic. Although adjusting for this bias only marginally reduces the ensemble-mean and spread of the projected SPV changes, it has pronounced implications for particular models at both ends of the range of projected SPV changes.

Overall, our study gives another example for a "tug-of-war" of different effects of global warming on atmospheric circulation changes. Understanding and quantifying these opposing effects is crucial to reduce uncertainties about regional climate change
scenarios.

**Data Availability**

Datasets for this research are available in these in-text data citation references and the associated repositories. Observational sea ice data (Titchner & Rayner, 2014): https://www.metoffice.gov.uk/hadobs/hadisst2/data/download.html; ERA5 reanalysis data (Hersbach et al., 2020): DOI: 10.24381/cds.6860a573; CMIP5 data (Taylor et al., 2012): https://esgf-node.llnl.gov/search/cmip5/


**Author contribution**
M.K. analysed the data and led the writing. All authors designed the study and contributed to the writing.





**Competing interests**

The authors declare no competing interest.

**Acknowledgments**

We thank ECMWF and the Hadley Centre for making the ERA5 and sea ice data publically available. We further acknowledge the World Climate Research Programme's Working Group on Coupled Modelling, which is responsible for CMIP, and we thank
the climate modeling groups listed in Fig 3 for producing and making available their model output. M.K. has received funding from the European Union's Horizon 2020 research and innovation programme under the Marie Skłodowska-Curie grant agreement [No 841902]. T.G.S. acknowledges support from ERC Advanced Grant 339390 (ACRCC).



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

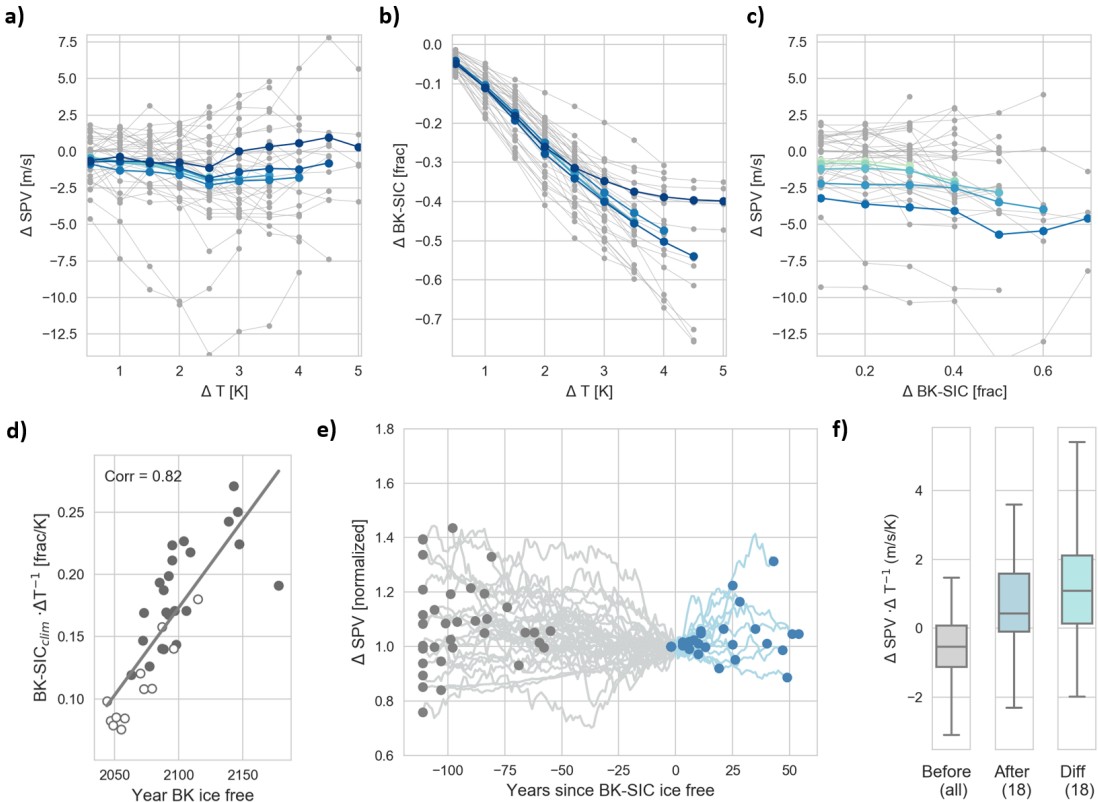


**Figure 1**. *Non-linear Response of the SPV to global warming.*

a) SPV change in winter (JFM) as a function of annual-mean global mean temperature change in the CMIP5 models (see methods). The thick blue lines show the ensemble mean for different sub-sets of reached warming level at the end of the 21st century, with darker colors indicating higher warming levels. For example, the darkest blue line shows the ensemble-

mean for the set of models reaching 5 K warming, while the second darkest line is the mean over all models reaching at least 4.5 K warming (thus including also those models reaching 5 K warming).  b) Same as a) but for BK-SIC change in autumn (OND) as a function of global mean temperature change.  c) Same as a) and b) but for SPV change as a function of BK-SIC change. d) Estimated timing of an ice-free BK in OND versus its climatology in 1960-1989 divided by global mean temperature change at the end of the 21st century, i.e. the warming averaged over the 2070-2099 period. The dark circles

indicate the models that have too much BK-SIC compared to observations, and the open circles those with too little sea ice. e) Time-series of moving 30-year mean SPV, normalized by the 30-year mean reached before the BK Seas become ice-free (in OND). Grey lines thus show evolution of change while there is sea ice, and blue lines for when BK is ice-free. Dots indicate the values at the end (blue) and at the start (grey). f) Boxplot of SPV change before and after BK is ice-free (normalized by global mean warming level over the respective period) as well as the difference for each model. The boxes

indicate the inter-quartile range and the whiskers represent the upper and lower quartile ranges. Only models which were ice-free for at least 10 years are included in the latter two boxplots (in total 18 models). In all panels, changes were calculated relative to the 1960-1989 reference period (see also methods).

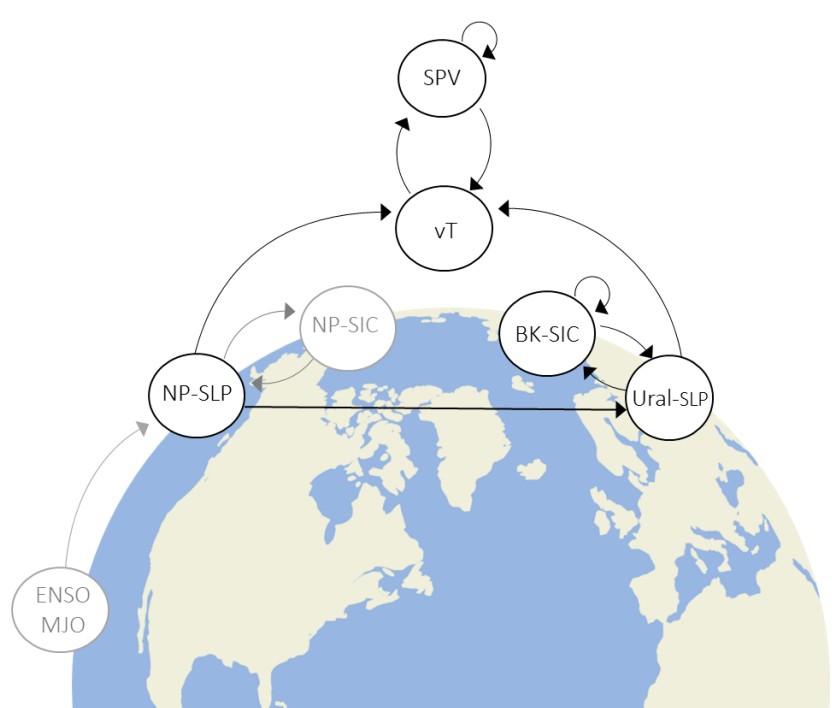


*Figure 2*. *Assumed causal model.*

Nodes in black contours represent the involved processes in the causal model: Barents and Kara sea ice concentrations ("BK-SIC"), sea level pressure over the Ural Mountains ("Ural-SLP") and over the North Pacific ("NP-SLP"), lower-stratospheric poleward eddy heat flux ("vT"), which is the upward wave forcing of the stratospheric circulation, and the stratospheric polar vortex ("SPV"). The black arrows represent the corresponding causal relationships, here assumed to operate on a monthly time-scale. The gray contoured nodes North Pacific sea ice concentrations ("NP-SIC") and El Niño Southern Oscillation/Madden Julian Oscillation ("ENSO/MJO"), and the respective arrows, denote processes discussed in the literature but not explicitly accounted for here because their effects are assumed to be mediated via NP-SLP.


**Figure 3.** *Individual links of Arctic-Stratosphere pathway.*

Links from a) seasonal-mean BK-SIC (in OND) to SPV (in JFM), and from b) monthly BK-SIC (in N) to Ural-SLP (in J), c) NP-SLP (in D) to Ural-SLP (in J), d) Ural-SLP (in N) to BK-SIC (in D), e) Ural-SLP (in D) to vT (in J), f) NP-SLP (in D) to vT (in J), and g) vT (in J) to SPV (in F).





Shown is the spread in link strength for each model (left panels), the distribution of all link strengths (middle panels) and of the models' means (right panels). The link strengths are quantified by regressing each variable on their parents (see Fig. 2) for each model over 39-year moving windows from 1900-2005 (in total 67 windows) in the historical simulations. Grey contours in the middle panel show the histogram obtained using (unadjusted) regression. The crosses in the middle panel denote the link strength obtained using observations (gray for unadjusted regression and colored crosses for regressions including all parents). Numbers in brackets after model names in x-axis indicate the used number of ensemble members, with no number meaning that just one member was used. The box and whisker plots thus include different amounts of data (number of ensemble members times the 67 moving windows). The boxes indicate the inter-quartile range, the whiskers represent the upper and lower quartile ranges, and horizontal lines show the median.

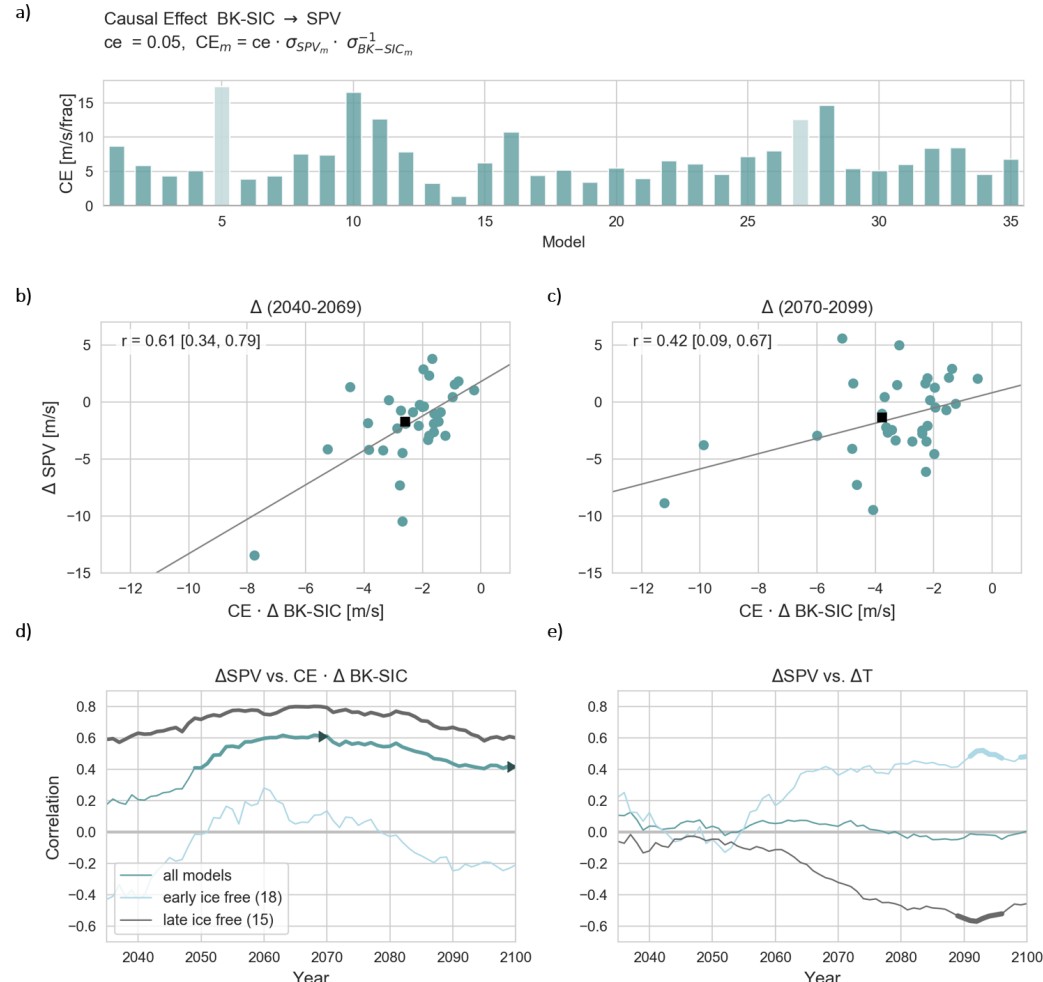


**Figure 4**. *Predicted vs Projected Polar Vortex Change.*

*a)* Causal Effect weighted by ratios of standard deviations of BK-SIC and SPV (calculated over the 1960-1989 reference period) to transform standardized ce = 0.05 into physical units. b) Projected vs. predicted winter (JFM) SPV change for mid-century (2040-2069). Prediction based on autumn (OND) sea ice. Each dot indicates one model. Changes are calculated relative to 1960-1989. Squares show ensemble means. r denotes the correlation coefficient and numbers in brackets the 95% confidence interval. c) Same as b) but for end-of-century (2070-2099) change. d) Green line shows the correlation (predicted vs projected SPV change) but for different moving windows. The years on the horizontal axis denote the last year of the 30-y average. Grey line shows the same but only for models for which BK-SIC is not ice free before 2090, blue line for models that are ice free before 2090. Thick parts of the lines indicate statistically significant correlation values (p<0.05) according to a two-sided Student´s t-test. e) Same as panel d) but for SPV change vs. global mean temperature change.



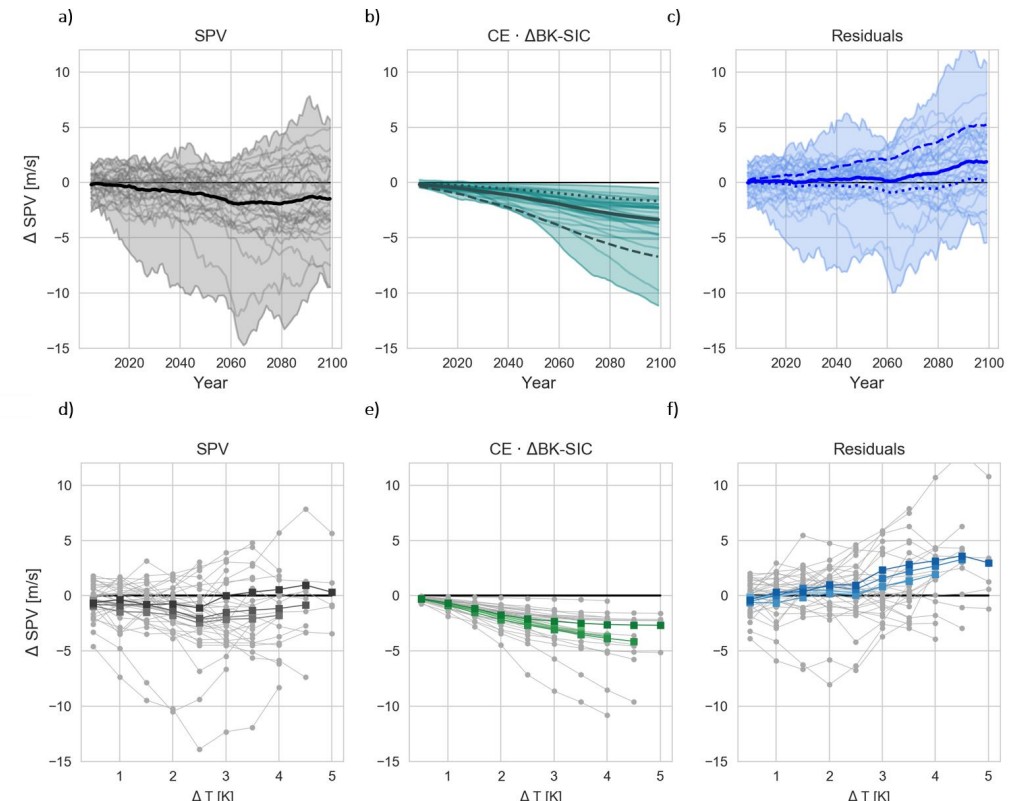

**Figure 5**. *Decomposed projected polar vortex response.*

30-year running mean of a) projected SPV change, b) BK-SIC-based predicted SPV change, and c) the residuals (b-a), as functions of time for a causal effect ce = 0.05. The thick lines indicate the ensemble mean and the dashed and dotted lines in panels b) and c) indicate the ensemble mean when using ce = 0.1 and 0.025 respectively. d), e), f) as a), b), c) but as a function of global mean temperature change and only for ce = 0.05. The thick lines show the ensemble mean for different sub-sets of reached warming level at the end of the 21st century, with darker colours indicating higher warming levels.

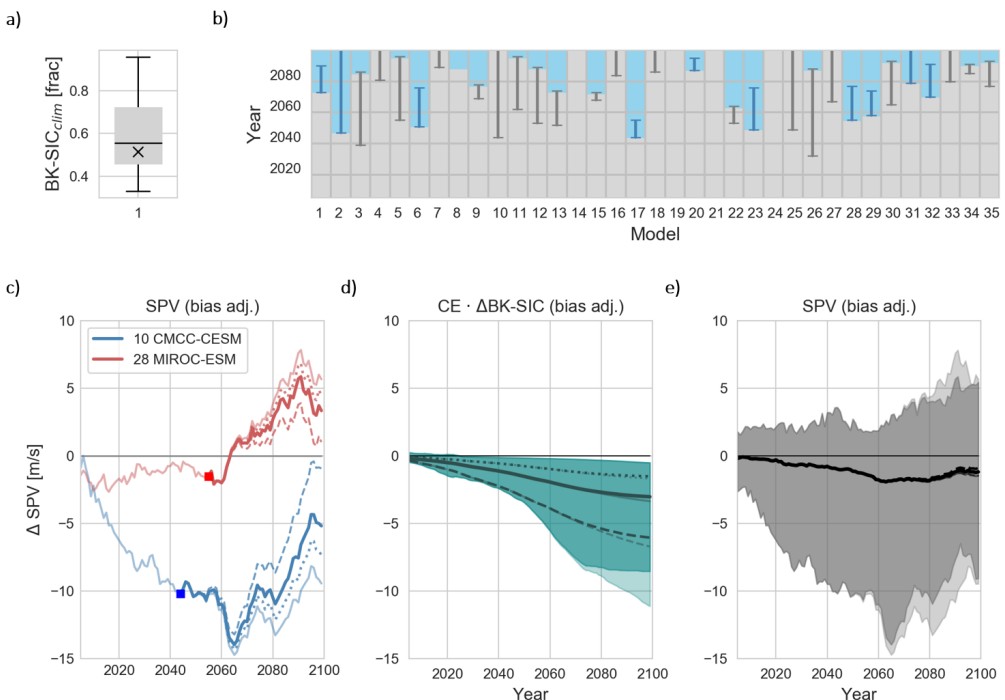

**Figure 6**. *Bias-adjusted polar vortex change.*

625     a) Spread across CMIP5 models of BK-SIC climatology (over 1960-1989 reference period). Black cross shows observational value and black line indicates the ensemble median. b) Grey shading indicates years when BK Seas are still ice covered, blue shading indicates years after the BK Seas have become ice free (see methods). Vertical lines indicate period over which the bias adjustment function ramps either up or down, grey for models with too much ice and blue for models with too little ice. c) Bias adjustment when assuming ce = 0.05 (thick lines) for models 10 and 28. Light thin lines

630     indicate the actual projected change, whilst dashed lines indicate bias adjustment when assuming a causal effect of ce = 0.1 and dotted lines that for ce = 0.025. d) Spread of predicted SPV from BK-SIC (light shading) and for bias adjusted BK-SIC for ce = 0.05 (dark shading). The dashed and dotted lines show the ensemble mean values for a causal effect of 0.1 and 0.025 respectively. e) Same as d) but for projected SPV and for visualization reasons not showing the values for ce = 0.025.

635