# Peer review of "The role of Barents-Kara sea ice loss in projected polar vortex changes"

_Weather and Climate Dynamics, 2020_

## Referee Comment (RC1) · Anonymous Referee #1 · 2 Aug 2020

Authors attempted to investigate the role of Barents-Kara (BK) sea ice loss in the future on stratospheric polar vortex (SPV) change. The BK sea ice loos at the moment appears to weaken the SPV on the observation (mainly based on reanalysis data), but whether this relation is due to global warming (i.e. forced response) or internal variability remains unclear. Some studies have suggested that the influence of the Arctic warming on mid-latitude weather may not exist or ends soon. Other studies suggested that a change in atmospheric circulation drives sea ice over the BK rather than the other way around. This study contributes to the clarification of the influence of BK sea ice loss on SPV and shows causal relation between the BK sea ice change and others. Though model responses are small to future loss of Sea ice and responds variously, ensemble mean suggests some meaning for the change in SPV, which has nonlinear

responses to the degree of global warming or BK sea ice loss. Overall, I recommend for publication in the journal WCD with minor revisions.

1. The relation between NP SLP to Ural SLP shows slight negative relation. What does this mean?

2. Currently, November BK SIC -> January Ural SLP is provides. How about other combinations like D BK SIC -> J Ural SLP, etc.

3. V'T' vs. BK SIC change or year (or degree of warming) would provide a good measure to determine why SPV has such nonlinear responses.

4. Models suggests that under large global warming, SPV will be enhanced. What would the possible factors for strengthening the future SPV.

5. Please fix line 246, Fig. 4c -> Fig. 4b, Fig. 4d-> Fig. 4c, and line 258, Fig. 4e - > Fig. 4d

---

## Referee Comment (RC2) · Anonymous Referee #2 · 26 Aug 2020

This study examines the potential impact of sea ice loss on the stratospheric polar vortex (SPV) in CMIP5 model projections. The authors find that the SPV shows a nonlinear response in the ensemble mean of CMIP5 projections. The SPV initially weakens slightly before showing strengthening again, resulting in little change over the 21st century. The authors show that this nonlinearity coincides with when the Barents-Kara (BK) Sea goes ice free, and thus present the hypothesis that BK sea ice is the cause of the nonlinearity. The authors then show that there is a very weak causal response of the SPV to fall BK sea ice in historical simulations, but this weak response can explain the initial SPV weakening. This weakening is then counteracted by other effects of global warming once the BK sea ice goes ice free.

The authors' hypothesis is plausible, and the study could compliment the large body of

literature that uses targeted sea ice loss experiments in models. However, there are a lot of issues that need to be addressed before it should be published.

Major issues:

1.The authors attribute the nonlinear response of the SPV (Fig 1a) in response to global warming to the BK sea ice. While this is certainly plausible, the evidence the authors present for this is not convincing. Fig 5 seems to show that a good portion (most?) of this nonlinear response actually comes from the residual term and not BK SIC. The change in the SPV is seen at around 2060 (Fig 5a) and 2.5K (Fig 5d) and this is clearly seen in the residual term (Fig 5c,f). The nonlinear response in the BK sea ice is a lot less obvious to me (Fig 5 b,e). How can the authors reconcile this result with their conclusions? The authors might want to consider quantifying how much the of nonlinearity is attributed to BKS SIC, perhaps by simply calculating the difference in linear trends before/after a certain year/temperature.

2. The authors' estimation of the causal effect likely overestimates the strength. While I commend the authors for trying the remove the effects of Ural SLP, I do not think what they have done fully accomplishes this. It is likely the Ural SLP in November and December that would impact both the JFM SPV and the OND BK sea ice. Peings (2019) showed that November Ural Blocking impacts both November BK sea ice and December-January SPV. Thus, averaging over SON will likely underestimate the co-founding effect because it includes September Ural SLP (which likely has little impact on the JFM SPV) but does not include December Ural SLP (which likely has a larger impacts on SPV and still impacts OND sea ice). I think removing the OND Ural SLP would be better. This could remove some of the impacts of sea ice on Ural SLP, but the authors show that this is likely to be small.

3. The authors show that there is a little causal effect of sea ice on Ural SLP, but that there is an effect on the SPV. How can the authors reconcile this with the causal model in Fig 2? Shouldn't there be a stronger causal effect between sea ice and Ural SLP

than with sea ice and the SPV? This seems to suggest that the causal effect on the SPV is overestimated (see above), or there is another pathway which BK sea ice can influence the SPV that does not involve Ural SLP.

4. The scatter plots/correlations from Fig 4 and the conclusions drawn from this analysis are flawed because they do not take into the cofounding effects, which likely exist. Because the same casual effect strength is used for all models, these plots/correlations would be identical if it was done with only the BK SIC (without the causal effect strength). We would expect these same correlations to arise if a change in Ural SLP (not caused by BK sea ice) affected both BK sea ice and the SPV. We would then expect this correlation to weaken in the later period because Ural SLP will have a weaker influence on BK SIC (because there is no/little sea ice remaining in some models).

5. From Fig 1a, it looks like there are a few outliers that have strong nonlinear response. A couple of these outliers are pointed out in Fig 6c. A few outliers also appear to have a big influence on the correlations in the scatter plots in Fig 4b,c. How much do these few outliers influence the model means? How many individual models display this nonlinearity? Individual models will be heavily influenced by internal variability so the authors might want to consider looking at available large ensembles (of which there now quite a few that are publicly available, see e.g. Deser et al. 2020) to see if nonlinearity is seen in individual models after removing internal variability.

Minor comments:

Title: I think it would be better to use 'Barents-Kara' instead of 'Arctic' because the study is almost entirely about the Barents-Kara sea ice.

L46: The weakening of the SPV in response to sea ice loss in modelling is not nearly as robust as portrayed here. Many studies find little impact on the SPV, but these tend to not highlight this result (negative results are not that exciting!). Some examples are: Semmler et al. (2020), Blackport and Screen (2019), Sun et al.( 2018). A weaker SPV in response to only BK sea ice loss might be more robust, but there are fewer studies

that have looked at this, and these are less relevant on climate change timescales (sea ice loss does not only occur in the BK Sea).

L48: "possible implications of future sea ice loss have so far been only rarely studied. . ." This is not true. Many of the studies referenced in the previous sentence (and many others) have used models to study the implications of future sea ice loss.

L51: None of these three studies looked at the SPV. Seviour (2017) and Garfinkel et al. (2017) might be better for this specific point.

L67: How sensitive are the results to the exact region used to define the BK Sea, especially for the timing of when BK goes ice free? A smaller region might go ice-free sooner and larger region will go ice free later. In terms of the potential forcing I do not think there anything special about the boundaries used here.

L153: Kug et al. 2015 did not show this. Many modelling studies find reduced SLP in the North Pacific in response to sea ice loss (e.g. Screen et al. 2018), so these could be better to cite here.

L170: How important is removing the effect of SON Ural SLP here (i.e. what is the value of b)?

L194: Although not the SPV specifically, this is supported by Kolstad and Screen (2019) who found the link between fall BK sea ice and winter NAO has been nonstationary over the last century and the recent period has been especially high.

L221: But this 'problem' exists with or without a role for sea ice (see major comment 1).

L271-275: I think it important to explicitly mention here (and possibly elsewhere in the paper), that the effects of global mean warming without BK sea ice loss included sea ice loss outside of the BK Sea. A number of studies have found that sea ice loss in the Pacific side of the Arctic causes a strengthening of the SPV (Sun et al. 2015; McKenna et al. 2017), so the effects of Arctic sea ice loss are likely even smaller than the effects

of BK sea ice loss.

L278: This should say BK SIC, not Arctic sea ice.

L326: These cited studies were not about the stratosphere, so are not that relevant here. Also, these studies were largely about refuting previous claims that there is evidence, and they are not claiming that the lack of evidence is evidence of no effect.

L327-328: Again, this not a very convincing argument, because BK SIC does not appear to explain a good portion of the nonlinearity. It is not completely unreasonable to think there could be non-linearity in the response of any of the many of the processes that drive the SPV, none of which were explored in this study.

L345: De and Wu, (2019) is not a good reference to make this point because it only correlations in preindustrial control simulations, not the response to sea ice loss. Also, this should be Blackport and Kushner 2017, although Screen et al. 2018 is a better reference for this point point because it used more CMIP5 models (including those used in Blackport and Kushner 2017).

Additional References:

Deser, C., and Coauthors, 2020: Insights from Earth system model initial-condition large ensembles and future prospects. Nature Climate Change, 10,277-286, https://doi.org/10.1038/s41558-020-0731-2.

Garfinkel, C. I., S.-W. Son, K. Song, V. Aquila, and L. D. Oman, 2017: Stratospheric variability contributed to and sustained the recent hiatus in Eurasian winter warming. Geophysical Research Letters, 44, 374–382, https://doi.org/10.1002/2016GL072035.

Semmler, T., F. Pithan, and T. Jung, 2020: Quantifying two-way influences between the Arctic and mid-latitudes through regionally increased CO2 concentrations in coupled climate simulations. Clim Dyn, 54, 3307–3321, https://doi.org/10.1007/s00382-020-05171-z.

Sun, L., M. Alexander, and C. Deser, 2018: Evolution of the Global Coupled Climate Response to Arctic Sea Ice Loss during 1990–2090 and Its Contribution to Climate Change. J. Climate, 31, 7823–7843, https://doi.org/10.1175/JCLI-D-18-0134.1.

---

## Author Comment (AC1) · 18 Sep 2020

Authors attempted to investigate the role of Barents-Kara (BK) sea ice loss in the future on stratospheric polar vortex (SPV) change. The BK sea ice loos at the moment appears to weaken the SPV on the observation (mainly based on reanalysis data), but whether this relation is due to global warming (i.e. forced response) or internal variability remains unclear. Some studies have suggested that the influence of the Arctic warming on mid-latitude weather may not exist or ends soon. Other studies suggested that a change in atmospheric circulation drives sea ice over the BK rather than the other way around. This study contributes to the clarification of the influence of BK sea ice loss on SPV and shows causal relation between the BK sea ice change and others. Though model responses are small to future loss of Sea ice and responds variously, ensemble mean suggests some meaning for the change in SPV, which has nonlinear responses to the degree of global warming or BK sea ice loss. Overall, I recommend for publication in the journal WCD with minor revisions.

We thank the reviewer for the overall very positive feedback.

1. The relation between NP SLP to Ural SLP shows slight negative relation. What does this mean?

We believe this slight negative relation reflects the presence of a link from the Aleutian Low to North Atlantic sea level pressure, via a Rossby wave train, as reported by several studies (e.g. Honda and Nakamura (2001)). Although NP-SLP turned out not to be too relevant on the seasonal time-scales (including it in the regression only marginally changes the ensemble-mean regression strength of BK-SIC to SPV from 0.052 to 0.048), we included it in the analysis as several previous studies showed an influence of tropical SST and Arctic sea ice variability on SPV strength via this pattern (e.g. Jimenez-Esteve & Domeisen (2018), McKenna et al. (2017)).

2. Currently, November BK SIC -> January Ural SLP is provides. How about other combinations like D BK SIC -> J Ural SLP, etc.

We agree with the reviewer that a link from November BK-SIC to January Ural-SLP is only one of several possible relationships which would all be consistent with the analysed seasonally averaged influence from OND BK-SIC to JFM SPV.  For the original submission we tested several such combinations covering different autumn/winter months and time-lags up to three months. All estimates provided a similar picture of a weak signal from BK-SIC to Ural-SLP on monthly time-scales. However, the signal in different models peaks at different lags, with the observations showing the most negative regression coefficient of -0.47 at lag-3 (in contrast, the observational lag-2 value presented in the manuscript is only -0.16), suggesting an important role of October BK-SIC for January Ural-SLP. However, here the scope was neither to optimize the potential signal from BK-SIC to Ural-SLP, nor to address potential model differences regarding the timing of this link. Therefore, given that we think there is no best choice (see also our response to the third comment of reviewer 2) we decided to show only one example here. Please also note that nothing in the subsequent analysis relies on this particular choice.

3. V'T' vs. BK SIC change or year (or degree of warming) would provide a good measure to determine why SPV has such nonlinear responses.

As suggested by the reviewer, we have calculated the change of vT as a function of ΔT and of ΔBK-SIC (see Fig. R1 below). We find that the ensemble-mean vT first increases (which is associated with a weakening of the SPV), peaking at a warming of 2.5 K and then plateaus with ongoing global-mean warming (see first panel in Fig. R1). Moreover, the vT change as a function of BK-SIC change shows an approximate linear increase. Thus, this analysis gives overall consistent results with the nonlinearity in SPV, and furthermore suggests that the nonlinearity originates in the troposphere rather than being associated with the SPV response. We have included these results in Figure 1 of the revised manuscript as it strengthens the overall evidence, and we thank the reviewer for making this suggestion.

[Figure]

Fig. R1. Same as Fig. 1a-c in the original manuscript but for changes in (DJF) vT instead of (JFM) SPV.

**4. Models suggests that under large global warming, SPV will be enhanced. What would the possible factors for strengthening the future SPV.**

This is an open question and is beyond the scope of our analysis, so we don't wish to speculate. In the discussion (l.333) we therefore simply state: *"Identifying the processes causing a strengthening of the SPV is beyond the scope of the present study but is relevant to understand future polar vortex change. Changes in both vertical as well as horizontal wave activity propagation might play a role (Wu et al., 2019). For example, a deepened Aleutian Low, as favoured by decreasing sea ice in the Pacific sector, might contribute to such a strengthening (Hu et al., 2018; McKenna et al., 2017; Nishii et al., 2010) (as suggested here in Fig. 3f)."*

**5. Please fix line 246, Fig. 4c -> Fig. 4b, Fig. 4d-> Fig. 4c, and line 258, Fig. 4e - >Fig. 4d**

Done.

References

Honda, M., and H. Nakamura (2001), Interannual seesaw between the Aleutian and Icelandic lows. Part II: Its significance in the interannual variability over the wintertime Northern Hemisphere, *J. Climate*, doi:10.1175/1520-0442(2001)014,4512:ISBTAA.2.0.CO;2.

Jiménez-Esteve, B., Domeisen, D. I. V., Jiménez-Esteve, B., & Domeisen, D. I. V. (2018). The Tropospheric Pathway of the ENSO–North Atlantic Teleconnection. *J. Climate*, 31(11), 4563–4584. doi: 10.1175/JCLI-D-17-0716.1

McKenna, C. M., Bracegirdle, T. J., Shuckburgh, E. F., Haynes, P. H., & Joshi, M. M. (2017). Arctic sea-ice loss in different regions leads to contrasting Northern Hemisphere impacts. *Geophysical Research Letters*. doi:10.1002/2017GL076433

**Anonymous Referee #2**

This study examines the potential impact of sea ice loss on the stratospheric polar vortex (SPV) in CMIP5 model projections. The authors find that the SPV shows a nonlinear response in the ensemble mean of CMIP5 projections. The SPV initially weakens slightly before showing strengthening again, resulting in little change over the 21st century. The authors show that this nonlinearity coincides with when the Barents-Kara (BK) Sea goes ice free, and thus present the hypothesis that BK sea ice is the cause of the nonlinearity. The authors then show that there is a very weak causal response of the SPV to fall BK sea ice in historical simulations, but this weak response can explain the initial SPV weakening. This weakening is then counteracted by other effects of global warming once the BK sea ice goes ice free. The authors' hypothesis is plausible, and the study could compliment the large body of literature that uses targeted sea ice loss experiments in models. However, there are a lot of issues that need to be addressed before it should be published.

We would like to thank the reviewer for taking the time to provide such detailed and constructive feedback which has helped us to reconsider the way we present our assumptions and results.

**Major issues:**

1. The authors attribute the nonlinear response of the SPV (Fig 1a) in response to global warming to the BK sea ice. While this is certainly plausible, the evidence the authors present for this is not convincing. Fig 5 seems to show that a good portion (most?) of this nonlinear response actually comes from the residual term and not BKSIC. The change in the SPV is seen at around 2060 (Fig 5a) and 2.5K (Fig 5d) and this is clearly seen in the residual term (Fig 5 c,f). The nonlinear response in the BK sea ice is a lot less obvious to me (Fig 5 b,e). How can the authors reconcile this result with their conclusions? The authors might want to consider quantifying how much the of nonlinearity is attributed to BKS SIC, perhaps by simply calculating the difference in linear trends before/after a certain year/temperature.

We thank the reviewer for making this comment but we think that it is at least partly subject to a misunderstanding. We make the argument for the nonlinear response of the SPV to BK-SIC on the basis of the evidence presented in Figure 1. Figure 5, instead, shows the *implications* of an assumed causal link from BK-SIC to SPV. More precisely, Fig. 5 has two purposes:

1) panels b and e show how even a small coupling between BK-SIC and SPV (relative to the year-to-year variability) implies a very substantial effect of BK-SIC loss on the SPV on long time scales, providing the evidence for one of the main conclusions of our study.

2) panels c and f show (via the residuals) the extent to which the nonlinearity of the SPV response can be modelled assuming a linear relation between BK-SIC and SPV, moreover one that is the same across all models --- both of which are clearly drastic simplifications of any true relationship.

In this respect, Figure 5 is not a test for nonlinearity, but of the extent to which the nonlinearity can be captured in such a simple way (which then provides the basis for the bias-adjustment performed to produce Figure 6, which supports another of the main conclusions of our study). In other words, in Figure 5 we are performing a process of *deduction* (not of induction as the reviewer appears to be assuming), which requires a different approach to the interpretation of the evidence. The best measure of the goodness of fit lies in the residuals, which indeed are not entirely linear in either year or global mean temperature, but in our assessment do not

exhibit deviations from linearity that are sufficiently strong (given all the simplifications involved, and the evident noise in the CMIP5 ensemble) to call into question the value of our subsequent bias-adjustment. Indeed, one could potentially use the goodness of fit to determine the coupling coefficient empirically (in panel c, taking ce = 0.1 would yield a close-to-linear trend in the residuals).

To prevent confusion about our reasoning, we have revised the manuscript and now explicitly state (l 227):

*"[..] our approach will now consist in exploring the implications for the SPV under climate change, assuming that a weak signal from BK-SIC to SPV indeed exists as suggested by previous studies (De & Wu, 2019; Kim et al., 2014; Screen, 2017b; Zhang et al., 2018) and supported by Fig. 3a. In other words, our approach is primarily one of deduction rather than of induction."*

**2. The authors' estimation of the causal effect likely overestimates the strength. While I commend the authors for trying the remove the effects of Ural SLP, I do not think what they have done fully accomplishes this. It is likely the Ural SLP in November and December that would impact both the JFM SPV and the OND BK sea ice. Peings (2019) showed that November Ural Blocking impacts both November BK sea ice and December-January SPV. Thus, averaging over SON will likely underestimate the confounding effect because it includes September Ural SLP (which likely has little impact on the JFM SPV) but does not include December Ural SLP (which likely has a larger impacts on SPV and still impacts OND sea ice). I think removing the OND Ural SLP would be better. This could remove some of the impacts of sea ice on Ural SLP, but the authors show that this is likely to be small.**

We certainly agree with the reviewer that our approach does not present an *exact* quantification of the causal effect of BK-SIC on SPV, or even what might be considered a best estimate. As we emphasized in the paper, uncertainties about time-scales and time-lags of the involved processes and the large internal atmospheric variability make the estimate noisy and overall highly uncertain. That is why our approach was, instead, to discuss *plausible* effect strengths and their implications. Nowhere do we claim to have accurately quantified the causal linkage. In particular, we would emphasize that our main results are presented in a qualitative manner; e.g. in the abstract we say:

*"we demonstrate that climate models show some partial support for the previously proposed link between low BK sea ice in autumn and a weakened winter SPV, but that this effect is plausibly very small relative to internal variability. Yet, given the expected dramatic decrease of sea ice in the future, a small causal effect can explain all of the projected ensemble-mean SPV weakening, approximately one-half of the ensemble spread at the middle of the 21st century, and one-third of the spread at the end of the century."*

For simplicity and motivated by the spread of the calculated model´s mean effects (see Fig. 3a) we therefore focused on average effect strengths of 0.025, 0.05, and 0.1. When including OND (instead of SON) Ural-SLP in the regression as suggested by the reviewer, the ensemble-mean estimate of the regression coefficients reduces from 0.052 to 0.036. This might reflect a weaker coupling, as the reviewer suggests, but in our opinion does not justify such a strong inference because of all the quantitative uncertainties associated with these calculations. (And with reference to the previous comment, such a weak coupling would imply more heteroskedasticity

in the residuals, and thus be a less plausible value from that perspective.) The key point in terms of our presented results is that this value lies within our plausible range.

As we believe that an exact quantification of the influence from BK-SIC to SPV is likely not feasible, we have kept the SON choice in the manuscript. However, in the revised manuscript we now also mention how the ensemble-mean effect changes when including OND Ural-SLP instead (l 190). We think that this also illustrates the benefit of our approach of presenting implications for different effect strengths, which allows the reader to bring in their own priors and assumptions to determine their own plausibility of these estimates.

**3. The authors show that there is a little causal effect of sea ice on Ural SLP, but that there is an effect on the SPV. How can the authors reconcile this with the causal model in Fig 2? Shouldn't there be a stronger causal effect between sea ice and Ural SLP with sea ice and the SPV? This seems to suggest that the causal effect on the SPV is overestimated (see above), or there is another pathway which BK sea ice can influence the SPV that does not involve Ural SLP.**

First of all, please note that the estimate of the effect of BK-SIC on SPV is based on seasonal-mean data while that of the link from BK-SIC to Ural-SLP uses monthly mean data, making them not directly comparable. Further, we interpret both estimates (see Fig. 3a and 3b) only as very rough estimates of the actual effect, motivating our approach to consider different plausible effect strengths (see also our response to comment 2 above). Given the large uncertainties about time-scales of interactions as well as potential non-stationarities (in particular regarding the two-way coupling of BK-SIC and Ural-SLP), we do not find it particularly surprising that our estimates of the strength of this mediating pathway are very noisy. Moreover, model differences in the spatial and temporal extent of the assumed pathway (something we did not address here) might contribute to this discrepancy. In fact, for most models, a much larger effect from BK-SIC to Ural-SLP can be found, if flexible choices of the time-lags between 1 to 3 months lag are allowed (not shown). We decided against such an approach as we felt that it would lead to a biased estimate, instead following a deductive approach assuming the presence of a weak link (see also our response to comment 4). We would also emphasize that none of our main conclusions, nor Figures 1 and 4-6, rely on the Ural pathway. Moreover, the main purpose of introducing the causal network in Figure 2 is not to determine the causal pathway, but to determine how to control for confounding factors based on the existing literature.

**4. The scatter plots/correlations from Fig 4 and the conclusions drawn from this analysis are flawed because they do not take into the cofounding effects, which likely exist. Because the same casual effect strength is used for all models, these plots/correlations would be identical if it was done with only the BK SIC (without the causal effect strength). We would expect these same correlations to arise if a change in Ural SLP (not caused by BK sea ice) affected both BK sea ice and the SPV. We would then expect this correlation to weaken in the later period because Ural SLP will have a weaker influence on BK SIC (because there is no/little sea ice remaining in some models).**

Note that we are fully aware of the point made in the second sentence of the reviewer's comment, which was stated explicitly in the original version of the manuscript.

As in our response to comment 1, we think there is a misunderstanding here. We do not make the argument for a causal link from BK-SIC to SPV based on the scatter plots/correlations in Figure 4. Rather, Figure 4 quantifies the implications of such a pathway (which is an active hypothesis in the scientific literature) on the time dependence of the SPV response to warming. In other words, it is a process of deduction, not of induction. In this context, Figure 4 supports the premise of BK-SIC influencing the SPV, but does not prove the premise. We tried to make this clear in the text by prefacing our statement about Figure 4 with *"Assuming a causal effect of 0.05"*.

As stated by the reviewer, an alternative interpretation of the data presented in Figure 4 could be that the statistical association of BK-SIC and SPV is solely due to a common driver Ural-SLP. However, as already mentioned at the beginning of Section 5, a common driver of the BK-SIC and SPV changes could not explain the non-linear SPV response seen in Figure 1, in the absence of any known mechanism that might do this.

Overall, we thus think that the confusion about Figure 4 arises from the deductive nature of our argument, which we therefore now stress explicitly in the revised version of the manuscript (l 230). Please also note that we explicitly decided against an estimate per model, given the before mentioned issues surrounding its calculation, as we were wary of overfitting. In fact, the correlations in Fig. 4 were somewhat higher when using model-specific estimates, but as mentioned before, we decided against this approach.

**5. From Fig 1a, it looks like there are a few outliers that have strong nonlinear response. A couple of these outliers are pointed out in Fig 6c. A few outliers also appear to have a big influence on the correlations in the scatter plots in Fig 4b,c. How much do these few outliers influence the model means? How many individual models display this nonlinearity? Individual models will be heavily influenced by internal variability so the authors might want to consider looking at available large ensembles (of which there now quite a few that are publicly available, see e.g. Deser et al. 2020) to see if nonlinearity is seen in individual models after removing internal variability.**

We have reproduced Figure 1a-c but indicating the multi-model *median* instead of the mean, as it is more robust to outliers (Fig. R2). Clearly, the non-linearity can still be seen, with the median indicating no further weakening after a warming of 2.5 K, which coincides with a flattening of BK-SIC in response to the global mean temperature rise. We also mention this now in the description of Fig. 1.

[Figure]

*Figure R2. Same as Fig. 1a-c of the original manuscript but with the blue lines indicating the multi-model median instead of the mean.*

Similarly, we also calculated the multi-model median (instead of the mean) in Figure 4b, c. Consistent with the findings, the prediction model explains 117% of the median change for the mid-century and 130% of the median change for the end-of century change. Please also note that we have used the median absolute deviation (MAD) to measure the ensemble spread, as this quantity is also less prone to outliers than the standard deviation. Hence, these more robust statistics also indicate a non-linear response of the SPV to global mean warming. Please also see our answer to comment to 3 of reviewer 1.

We definitely agree with the reviewer that investigating large single-model ensembles will be key to address the nonlinearity in more detail. However, we would like to stress that a non-linear SPV response was detected in the large-ensemble study using the MPI model (Manzini et al. (2018)), and found to coincide with the BK becoming ice-free, consistent with the results presented here. Large single-model ensembles might also help to understand potential non-stationarities affecting the SPV response to BK-SIC decrease (for example, Labe et al. (2019) showed that the phase of the QBO modulates the Ural-SLP and also the SPV response to sea ice loss). However, these aspects are beyond the scope of our study.

References:

Manzini, E., Karpechko, A. Y., & Kornblueh, L. (2018). Nonlinear Response of the Stratosphere and the North Atlantic-European Climate to Global Warming. *Geophysical Research Letters*, doi:10.1029/2018GL077826

Labe, Z.M., Y. Peings, and G. Magnusdottir (2019). The effect of QBO phase on the atmospheric response to projected Arctic sea ice loss in early winter, *Geophysical Research Letters*, doi:10.1029/2019GL083095

**Minor comments:**

Title: I think it would be better to use 'Barents-Kara' instead of 'Arctic' because the study is almost entirely about the Barents-Kara sea ice.

*We agree and have changed the title accordingly.*

L46: The weakening of the SPV in response to sea ice loss in modelling is not nearly as robust as portrayed here. Many studies find little impact on the SPV, but these tend to not highlight this result (negative results are not that exciting!). Some examples are: Semmler et al. (2020), Blackport and Screen (2019), Sun et al. (2018). A weaker SPV in response to only BK sea ice loss might be more robust, but there are fewer studies that have looked at this, and these are less relevant on climate change timescales (sea ice loss does not only occur in the BK Sea).

*We removed the work "robustly". The sentence now reads: "[...] the potential of decreasing BK-SIC to weaken the SPV on longer time-scales has been shown in various targeted model experiments."*

*Please also note that we only refer to the BK here.*

L48: "possible implications of future sea ice loss have so far been only rarely studied..."This is not true. Many of the studies referenced in the previous sentence (and many others) have used models to study the implications of future sea ice loss.

*We have changed this sentence (l 49): "In what way future BK sea ice loss will affect the SPV is, however, not clear (McKenna et al., 2017; Sun et al., 2015)."*

L51: None of these three studies looked at the SPV. Seviour (2017) and Garfinkel et al. (2017) might be better for this specific point.

*Thank you. We have changed the references.*

L67: How sensitive are the results to the exact region used to define the BK Sea, especially for the timing of when BK goes ice free? A smaller region might go ice-free sooner and larger region will go ice free later. In terms of the potential forcing I do not think there anything special about the boundaries used here.

*We did not test this explicitly but we used (following Screen et al. (2017a)) a relatively large region to describe BK-SIC (65-85°N and 10-100°E).*

L153: Kug et al. 2015 did not show this. Many modelling studies find reduced SLP in the North Pacific in response to sea ice loss (e.g. Screen et al. 2018), so these could be better to cite here.

*Done.*

L170: How important is removing the effect of SON Ural SLP here (i.e. what is the value of b)?

The effect is small with the ensemble-mean (of the model´s means) of the regression coefficient being close to zero (see also Fig. R3 below).

[Figure]

*Figure R3. Box and whiskers plot of the model´s means spread of regression parameter a (left panel) and b (right panel).*

**L194: Although not the SPV specifically, this is supported by Kolstad and Screen (2019) who found the link between fall BK sea ice and winter NAO has been nonstationary over the last century and the recent period has been especially high.**

We cite this study in l 167.

**L221: But this 'problem' exists with or without a role for sea ice (see major comment1).**

Please see our answer to comment 1.

**L271-275: I think it important to explicitly mention here (and possibly elsewhere in the paper), that the effects of global mean warming without BK sea ice loss included sea ice loss outside of the BK Sea. A number of studies have found that sea ice loss in the Pacific side of the Arctic causes a strengthening of the SPV (Sun et al. 2015; McKenna et al. 2017), so the effects of Arctic sea ice loss are likely even smaller than the effects of BK sea ice loss.**

We agree and have now state this in the manuscript (l 293): *"Note that these other effects could also include potential effects of Arctic sea ice loss in regions other than the BK Seas (McKenna et al., 2017; Sun et al., 2015)."*

**L278: This should say BK SIC, not Arctic sea ice.**

Done.

**L326: These cited studies were not about the stratosphere, so are not that relevant here. Also, these studies were largely about refuting previous claims that there is evidence, and they are not claiming that the lack of evidence is evidence of no effect.**

Note that this paragraph (including the first sentence) is more generally about potential Arctic-midlatitude linkages. We have change the sentence, it now says (l 330):

*"Several previous studies have stressed the absence of a statistically significant signal to question claims concerning the influence of sea ice (Blackport and Screen, 2020; Seviour, 2017; Sun et al., 2016). However, absence of evidence is not evidence of absence (Shepherd, 2016)."*

L327-328: Again, this not a very convincing argument, because BK SIC does not appear to explain a good portion of the nonlinearity. It is not completely unreasonable to think there could be non-linearity in the response of any of the many of the processes that drive the SPV, none of which were explored in this study.

It may not be completely unreasonable, but we are not aware of a single published study that makes such a case. The only point we make here is that from a Bayesian perspective it is important to consider the implications of the hypothesis of no causal link as well as that of a causal link.

L345: De and Wu, (2019) is not a good reference to make this point because it only correlations in preindustrial control simulations, not the response to sea ice loss. Also, this should be Blackport and Kushner 2017, although Screen et al. 2018 is a better reference for this point point because it used more CMIP5 models (including those used in Blackport and Kushner 2017).

We have changed the references.

Additional References:

Deser, C., and Coauthors, 2020: Insights from Earth system model initial-conditionlarge ensembles and future prospects .Nature Climate Change, 10,277-286, https://doi.org/10.1038/s41558-020-0731-2.

Garfinkel, C. I., S.-W. Son, K. Song, V. Aquila, and L. D. Oman, 2017: Stratospheric variability contributed to and sustained the recent hiatus in Eurasian winter warming. Geophysical Research Letters, 44, 374–382, https://doi.org/10.1002/2016GL072035.

Semmler, T., F. Pithan, and T. Jung, 2020: Quantifying two-way influences between the Arctic and mid-latitudes through regionally increased CO2 concentrations in coupled climate simulations. Clim Dyn, 54, 3307–3321, https://doi.org/10.1007/s00382-020-05171-z.C5

Sun, L., M. Alexander, and C. Deser, 2018: Evolution of the Global Coupled Climate Response to Arctic Sea Ice Loss during 1990–2090 and Its Contribution to Climate Change. J. Climate, 31, 7823–7843, https://doi.org/10.1175/JCLI-D-18-0134.1.

---

## Author Response (AR2)

Major Comment

The authors have improved the manuscript, but the authors did not address my first major comment about the nonlinear response of the SPV and I am very confused by their response. It is not clear to me how framing it as a deduction rather than induction helps to clarify anything. I agree with their point 1) in their response and this is an important implication of the analysis. However, another deduction/implication of the assumed causal link from Fig 5 is that BK-SIC loss cannot explain the nonlinearity of the SPV, which contradicts one of the main conclusions of the paper. The authors state that Fig 5 shows that a weak causal influence of BK-SIC can explain the nonlinearity of the SPV (L337), but I don't think Fig 5 shows this at all. In the last few sentences of their response to this point, the authors state that they think the trend in the residual is linear. I disagree with this. To me, it looks like most of the nonlinearity in the SPV response (Fig 5a,d) actually comes from the residual (Fig 5c,f) not the BK SIC (Fig 5 b,e). Maybe my eyes are deceiving me, which is why I suggested the authors do something more quantitative, so we do not have to debate about what it looks like.

We thank the reviewer for his/her comments, which motivated us to provide more analysis to support our conclusions, as discussed below in more detail. We have split the reviewer´s comment, as well as our response, into two parts. The first part of our response relates to the interpretation of Fig 1, and the second part to the interpretation of Fig. 5 (now Fig. 6). However, we would also note that our statement of the role of BK-SIC in the nonlinearity of the SPV response was already quite weak; e.g. in the Abstract we said: „*We provide evidence for a non-linear response of the SPV to global mean temperature change, coincident with the time the BK Seas become ice-free.*"

[Part 1: regarding the interpretation of Fig 1]

In their response, the authors also state that the evidence for nonlinear response to BK-SIC is presented in Figure 1. However, the nonlinearity in the BK-SIC is primarily seen in the models with >5K warming and extremely weak when looking at all models (Fig 1c), but the nonlinearity of the SPV is apparent in all models, not only the >5K models (Fig 1a). This again appears to contradict the conclusion that the nonlinearity is driven by BK-SIC.

First please note that it makes sense that the signal of nonlinearity is stronger for the models with >5 K warming because they will be ice-free for longer, whereas the models that warm less will be ice-free for less time, which means that nonlinearity will be less evident, given the noise.

To provide additional evidence for the alleged non-linear SPV response related to BK-SIC loss, we computed Fig. 1 a-c but with the multi-model mean indicated separately for the subsets of early (before the year 2090) and late (after the year 2090) ice-free models. As expected, there is a non-linearity in the SPV, vT and BK-SIC response to global mean warming for the former group of models. In contrast, the latter group shows an approximately linear response, including in particular an (sub-)ensemble mean weakening of the SPV. Thus, the non-linearity seen in Fig. 1a-c comes from the non-linearity of the early ice-free models, supporting our hypothesis. In the revised version of the manuscript, we have included these panels as a new figure 2 of the manuscript (see below), together with the former panels of Fig. 1f, g.

Furthermore, note that for 14 out of the 18 early ice-free models, the difference of the SPV change before and after the BK has become ice-free is positive (see turquoise box and whiskers plot in new Fig. 2e, below). When assuming no role of BK-SIC, the probability of this occurring by chance is only 1 %, following a binomial distribution. This result, together with Fig. 1 and Fig. 2a-c, are in our opinion overall convincing evidence for a non-linear response of the SPV dependent on the timing of the BK becoming ice-free.

[Figure]

*New Figure 2. a)-c) as In Fig. 1 a-c but with the ensemble-mean calculated separately for the early ice-free models (in blue) and late ice-free models (in dark grey). d) and e) as before in the former Fig. 1*

[Part 2: regarding the interpretation of Fig 5 (now Fig 6)]

The authors then use this nonlinearity of the SPV as evidence for the causal link between BK-SIC and SPV and claim it is a 'problem' or a 'puzzle' to explain the nonlinear response if it is not sea ice (L226-227, L332-334). However, based on the evidence presented, a small causal link can't explain it either, so this is a problem or puzzle either way. In their response, the authors say that there is no published study that makes the case for a nonlinear response from other processes. I am not aware of any other study that has shown the nonlinearity in the SPV response, so of course there is no evidence that other processes are to blame. Also, the amount of nonlinearity in the SPV response is quite small, which makes it more difficult to rule out other processes.

The purpose of Figure 5 (now Figure 6) was to show the implications of the hypothesized causal link from BK-SIC to SPV. The reviewer is, however, right that also for this deductive exercise it is absolutely reasonable to ask whether the calculations provide evidence *against* our hypothesis. In other words, does the data (i.e. the CMIP5 RCP8.5 projections of SPV, BK-SIC and T), give reason to believe that the alternative hypothesis H1: "BK-SIC is no source of non-linearity of future SPV changes" is more likely than our initial assumption H0: "BK-SIC is a source of non-linearity (with a causal effect ranging between 0.025 and 0.1)".

One can address this by comparing the probability of seeing the data under H1, with that of the probability of seeing it under H0. This ratio is also known as the Bayes Factor (BF), and is used in Bayesian hypothesis testing:

$$BF = P(data \mid H1) / P(data \mid H0)$$

To calculate BF, we first translate the two hypotheses into two linear regression models

$$H0: \Delta SPV = ce\, \Delta BK\text{-}SIC + b_0\, \Delta T + \varepsilon_0 \qquad \text{with ce in } [0.025, 0.1]$$

$$H1: \Delta SPV = b_1\, \Delta T + \varepsilon_1$$

and determine $b_0$ and $b_1$ for each climate model by regressing $\Delta$SPV on $\Delta$T (to estimate $b_1$) and by regressing ($\Delta$SPV – ce $\Delta$BK-SIC) on $\Delta$T (to estimate $b_0$), whereby we also iterate over different ce for the latter (which assumes equal prior probability of all ce within the specified range). The Bayes factor thus quantitatively describes if the SPV changes as presented in Figure 6d are "more linear" than the Residuals in Fig. 6f, where the effect of sea ice has been removed in the latter. We consider this to be the quantitative assessment that the reviewer has asked for, rather than relying on subjective judgements based on how things look by eye.

It can be shown that for Gaussian noise $\varepsilon_0$ and $\varepsilon_1$, the probabilities P(data | H0) and P(data | H1) are also Gaussian and can thus be easily computed. They only depend on the squared errors (observed minus predicted values) of H0 and H1 as well as on the variance of the noise terms. We estimate the latter as the SPV variance over the historical runs (see also Supplementary Material for a more detailed description).

The different regression plots and the Bayes Factor are shown below (Fig. S1). The key result here is, that for almost all CMIP5 models, the Bayes Factor is close to 1, meaning that the data are similarly likely under both hypotheses. Only for Model 11 (BF = 20.7), model 14 (BF = 4.7) and model 23 (BF = 4.4) does the data appear more likely under H1. However, note that for these models the errors for H0 are actually also small. In summary, the CMIP5 projections thus do not provide evidence against our H0, and therefore justify keeping our deductive approach on the basis of Fig. 1 and Fig. 2.

In the revised version of the manuscript we have included a brief statement on this new analyses (l 289-298) and provide a more detailed discussion in the Supplementary Material which also includes Figure S1. Moreover, we removed the statement in the conclusion section which claimed that Fig. 5 explains the non-linearity in the non-linearity in the SPV response. We believe that these qualitative and quantitative changes have greatly improved this part of the paper and we would like to thank the reviewer again for challenging us to do so.

[Figure]

Fig. S1. Bayes factor analysis.

Shown are for each CMIP5 model, ΔSPV as a function of ΔT (grey dots, as in Fig. 6d) as well as the Residuals, i.e. (ΔSPV − ce BK-SIC), for the most extreme case of ce = 0.1 (blue dots; similar to Fig. 6f, but note that the Residuals in Fig. 6f are shown for ce = 0.05). The lines show $b_0$ ΔT (in blue) and $b_1$ ΔT (in grey). The Bayes factor (BF) of each model as well as the year the BK Seas become ice-free (if) are indicated in each panel.

Overall, there a still some nice work in this study, so I still think it should be published. The causal analysis (Fig 3) and the implications that even a very small causal effect could still have an impact on the response and model spread of the SPV is an important result. However, the evidence presented by the authors does not support a link between nonlinearity of the SPV response and BK-SIC. The authors should have less focus on this result/conclusion, or provide additional evidence for it.

We are glad the reviewer appreciates our findings and we hope that the additional analysis makes our work even more convincing.

Minor Comment

L47-49: If the authors are only referring to BK sea ice here (as stated in their response), then Blackport and Kushner 2017, Hoshi et al. 2017, Nakamura et al. 2016 and Screen 2017b, should be removed as references here. These studies look at the response to either observed or projected Pan-Arctic sea ice loss (not just BK).

[revised manuscript text omitted]